# Effects of virtual reality's viewing medium and the environment's spatial openness on divergent thinking

**Kenshiro Ichimura**⊙*

Research Department, National Institution for Academic Degrees and Quality Enhancement of Higher Education, Kodaira-shi, Tokyo, Japan

* itimura-k@niad.ac.jp

## Abstract

Virtual Reality (VR) technology is used in various fields, and research on VR creative activities has been widely conducted. This study examined the effects of VR environment on divergent thinking, a component of creative thinking. Specifically, two experiments were conducted to test the prediction that viewing visually open VR environments with immersive head-mounted displays (HMD) affects divergent thinking. Divergent thinking was evaluated using Alternative Uses Test (AUT) scores; AUT was performed while the participants viewed the experiment stimuli. In Experiment 1, I manipulated the VR viewing medium by having one group view a 360˚ video with an HMD and a second group view the same video on a computer screen. Additionally, I established a control group that viewed a real-world laboratory instead of the videos. The HMD group showed higher AUT scores than the computer screen group. In Experiment 2, I manipulated the spatial openness of a VR environment by having one group view a 360˚ video of a visually open coast and a second group view a 360˚ video of a visually closed laboratory. The coast group showed higher AUT scores than the laboratory group. In conclusion, exposure to a visually open VR environment on an HMD promotes divergent thinking. The limitations of this study and suggestions for further research are discussed.

## Introduction

Virtual reality (VR) technology has been used in various fields, such as consumer activities, medical care, sports, entertainment, and education [1–7]. In particular, the immersive capability of various spaces' re-creation in VR is considered suitable for creative activities. Most VR studies on creative activities have examined VR technologies and training programs that promote such activities [8–10], but not the specific elements that enhance creativity. Therefore, the present study aimed to identify the factors of VR that affect creativity.

Creativity and creative cognition are complex, and many perspectives on the topic have minimal consensus [11, 12]. However, in the current study, I consider creativity based on the perspective of Guildford [13]. Guildford [13] assumes that creative cognition consists of two

**Funding:** This work was supported by JSPS KAKENHI (grant number 20K14158). The funders had no role in study design, data collection and

analysis, decision to publish, or preparation of the manuscript.

**Competing interests:** The author has declared that no competing interests exist.

major thought processes: divergent and convergent thinking. Divergent thinking is the ability to generate various ideas in open-ended problems by making unexpected combinations, recognizing links among remote associates, and transforming information into unforeseen forms [14]. Meanwhile, convergent thinking is the ability to produce a single best answer to clearly defined questions by applying conventional and logical search, recognition, and decision-making strategies [15].

The present study focused on divergent thinking because of its affinity with VR. Divergent thinking is an essential skill to deal with the various problems of today's increasingly complex and diverse society [16]. Exploratory experiences and free concept activation promote divergent thinking, but providing such experiences in schools and workplaces is difficult. In contrast, VR can simulate such special situations with relative ease [8, 17, 18]. Studies have focused on this feature of VR and demonstrated that VR experiences promote divergent thinking [9, 19]. However, they have not examined which VR factors influence divergent thinking. Studies have indicated that immersion in the environment and different spaces affect divergent thinking in real-world settings [20, 21]. Therefore, by clarifying the effects of immersion and spatial differences on divergent thinking, we can offer suggestions for introducing VR more effectively into creative activities.

## Immersion and divergent thinking

Researchers have examined the effects of immersion on creativity in real-world and VR settings [20, 22, 23]. As an example that focused on divergent thinking, Lynch et al. [20] conducted an experiment with participants between the ages of 9 and 14 at a two-week residential summer camp. They found that immersion in a natural environment, away from media and technology throughout the camp, resulted in higher performance on a divergent thinking task. Palanica et al. [23] conducted an experiment with adults to compare the effects of natural and urban environments on divergent thinking at different levels of psychological immersion. Their results showed that the divergent thinking task performance was higher when viewing nature videos compared to urban videos in 2D and 3D VR. Conversely, performance was higher in both natural and urban environments in real-world settings, which were considered the most immersive.

These results suggest that divergent thinking tends to be higher when people are more immersed in the environment. However, since it is difficult to directly manipulate the degree of immersion in an environment in real-world settings; thus, it is unclear whether the creativity-enhancing effect is due to increased exposure to the environment or other factors [22]. In contrast, VR settings can directly manipulate the degree of immersion. For instance, displaying environments in 3D VR formats can enhance psychological immersion compared to 2D video formats [18]. In particular, immersive head-mounted displays (HMD), which are widely used for home use, have the effect of enhancing visual presence by providing visual information at a highly realistic level; although, the degree of immersion is less than that of systems that reproduce multiple senses [24]. Therefore, if immersion in the environment is associated with divergent thinking, VR experiences using an HMD will elicit higher divergent thinking performances than those on computer screens. This hypothesis is inconsistent with the results of Palanica et al.'s [23] study, which found no main effect or interaction of medium (2D videos vs. 3D VR). However, in their experiment, participants viewed videos for 45 seconds prior to the task and for three minutes while performing the task, which may have been too short a duration for adequate immersion in the environment. Therefore, there is room to extend the video viewing time and re-examine the differences between 2D and 3D VR. Experiment 1 of

the present study compared the effects of viewing 360˚ videos on an HMD versus on a computer screen.

## Environment and divergent thinking

Environmental differences have been reported to affect creativity in real-world settings [21, 25–30]. As for divergent thinking, Vohs et al. [21] compared orderly and disorderly environments and revealed that disorderly rooms enhanced students' divergent thinking. Chan and Nokes-Malach [25] tested the effects of space size on adults' divergent thinking process and showed that larger rooms produce more novel ideas than smaller rooms. Meyers-Levy and Zhu [29] examined the effect of ceiling height on the type of information processing and found that higher ceilings promote students' divergent thinking compared to lower ceilings. The effect of spatial differences on divergent thinking is also examined in VR settings. Palanica et al. [23] showed that viewing nature in 2D videos or 3D VR led to higher performance on a divergent thinking task than viewing an urban environment (although in real-world settings, the performance was higher in both nature and cities). Guegan et al. [26] compared the VR settings of a headmistress's office, a schoolyard, and a dreamlike environment and found that the latter two promoted children's divergent thinking.

The environments that promote divergent thinking may have aspects that activate the concept of freedom. It has been previously argued that there might be a relationship between environment and concept activation. For example, Hall [31] argued that small and contained spaces could prime notions of confinement or restrictedness, whereas larger spaces can prime notions of freedom and openness. Moore et al. [32] suggested that lower ceilings may invoke more restricted play, whereas higher ceilings may encourage freer play. In addition, Meyers-Levy and Zhu [29] theorized that high ceilings activate freedom-related concepts and then induce relational information processing, whereas low ceilings activate confinement-related concepts and induce item-specific processing. Similarly, Steidle and Werth [30] showed that dim lighting enhances creativity task performance, explaining that darkness offers freedom from constraints.

Thus, larger rooms, higher ceilings, disorderly rooms, nature, schoolyards, and dreamlike environments, which are shown to promote divergent thinking, are more likely to activate free concepts than smaller rooms, lower ceilings, orderly rooms, urban settings, and a headmistress's office. Therefore, providing a VR environment that can activate the concept of freedom may positively affect divergent thinking. When we want to engage in divergent thinking, we do not necessarily have access to such an environment in the real world, but we can easily access such an environment in VR. Therefore, studying the effects of different environments in VR holds practical value. To clarify environments' effects on divergent thinking, Experiment 2 compared a video of a visually open coast with no ceilings, walls, or objects blocking the view, which is an environment conducive to activating the concept of freedom, and a video of a visually closed laboratory.

## Overview of the experiments

As established, the VR environment, which can elicit a high level of immersion and activate the concept of freedom, is suitable to promote divergent thinking. This study conducted two experiments to test the effects of VR on divergent thinking. However, it is difficult to isolate the effects of these two factors (i.e., level of immersion and spatial openness) in two experiments, because confounding factors affecting divergent thinking are inevitable. Therefore, as a starting point to examine the effects of VR environments on divergent thinking, this study compared conditions thought to have markedly different effects on divergent thinking and

examined whether the predicted results were obtained. I then discuss other considerable factors and ideas for future research. This study aimed to provide insights into the relationship between VR environments and divergent thinking.

Across two experiments, the Alternative Uses Test (AUT) [33] was used to measure divergent thinking. In AUT, participants generate as many ideas as possible on alternative uses for everyday objects (e.g., bricks and paper clips). Subsequently, the fluency (number of ideas generated), originality (novelty of the generated ideas), flexibility (range of ideas in different categories), and elaboration (level of details the ideas contain) of the ideas are evaluated as measures of divergent thinking. AUT is a commonly used divergent thinking task, and several studies have confirmed its reliability and validity [34–39].

In Experiment 1, I manipulated the VR viewing medium to test the effect of an immersive HMD. Specifically, I established two groups: one group viewed a 360˚ video of a visually open coast on a computer screen (low reality). The other group viewed the same video on an HMD (high reality). In addition, I established a control group that viewed a real-world laboratory instead of the open coast video. The control group was assumed to be the baseline AUT performance, because they remained unexposed to the video stimuli. I predicted that the AUT scores would be higher in the HMD group than in the computer screen group (Prediction 1) because HMDs offer a more real and immersive visual environment than computer screens.

In Experiment 2, I manipulated the spatial openness of the VR environment. Specifically, I established a group that viewed a 360˚ video of the coast (open environment) on an HMD and a second group that viewed a 360˚ video of the laboratory (closed environment) on an HMD. I predicted that the AUT scores would be higher in the coast group than in the laboratory group (Prediction 2), because the open coast VR environment is more likely to activate the concept of freedom than the closed laboratory VR environment.

## Experiment 1

Experiment 1 investigated the effect of the VR experience with an HMD on divergent thinking. For this purpose, I compared the AUT scores of the group that viewed the 360˚ video of the coast on an HMD with the AUT scores of the group that viewed the same video on a computer screen.

### Methods

**Participants.** I conducted a statistical power analysis for sample size estimation using G*Power 3.1. With reference to the effect sizes reported in previous studies examining the effects of different physical environments on divergent thinking [21, 25, 30], I assumed the effect size in the present study to be large ($f = .40$) using Cohen's [40] criterion. With alpha = .05 and power = .80, the projected sample size needed with this effect size was $N = 66$. To meet this sample size, I recruited 75 university students to participate in Experiment 1 (54 males; $M_{age} = 21.6$ years, $SD_{age} = 2.8$). They were recruited through on-campus advertisements and assigned to one of the three groups while ensuring equal gender ratios (HMD, computer screen, and control; $n = 25$). All participants signed informed written consent forms prior to beginning the experiment and were compensated JPY 1,000 after the experiment for their participation. This experiment was approved by the ethics review committee of the institution at which the experiments were conducted.

**Stimuli and apparatus.** I used a 360˚ video of the Okinawan coast (https://360rtc.com/videos/tennohamabeach001/) as the stimulus (Fig 1A). The video was captured from a vantage point with few people or objects blocking the view. The video was 10 seconds long and played on a loop. The HMD group viewed the video on a headset (Oculus Go), and the computer

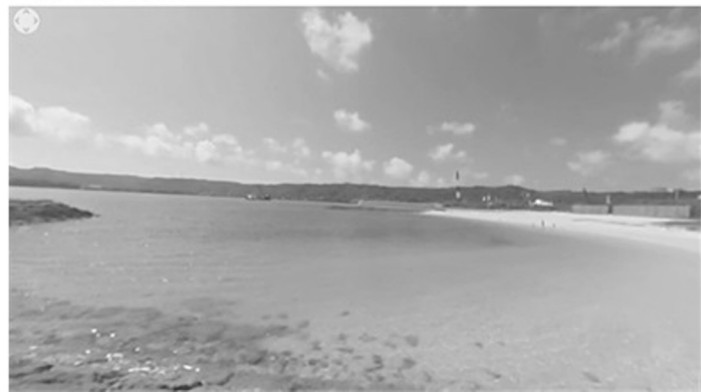 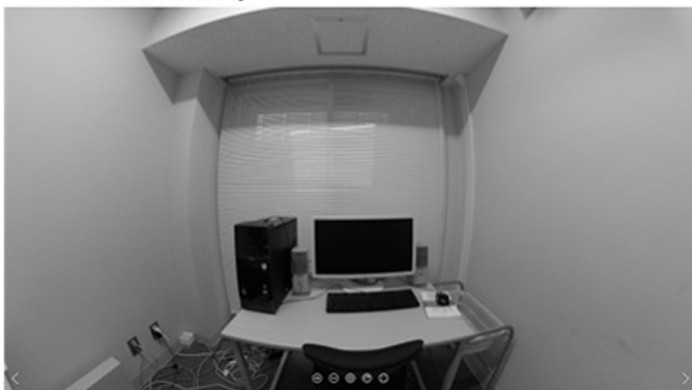

**Fig 1.** View of (a) the 360˚ video of the coast and (b) the laboratory.

screen group viewed it on a 23.6-inch monitor (iiyama PLB2409HDS). The control group did not view the video; they viewed the real-world laboratory (2.5 × 3.5 m in size) from the center of the room (Fig 1B). The sound of the video was muted to control only the visual information, and the laboratory was soundproofed to maintain tranquility.

**Procedure.** The experiment was conducted in a laboratory, as shown in Fig 1B. Initially, participants viewed the stimuli for eight minutes. The HMD group wore the headset and viewed the video freely at 360˚ while standing in the center of the room. The computer screen group sat at a desk and viewed the video on a computer monitor while moving their viewpoint freely at 360˚ with a mouse. The control group looked around the laboratory freely while standing in the center of the room. After a three minutes break, participants answered the AUT while viewing the video on an HMD or computer screen or looking around the laboratory. As part of the AUT, participants were presented with eight objects: a brick, newspaper, ping-pong ball, wooden board, hanger, cup, sock, and paper clip. The researcher presented the objects orally, one at a time, and the participants answered orally. The response time for each object was two minutes. The participants' responses were audio recorded. The entire experiment took approximately 40 minutes.

**AUT scoring.** Three coders, blinded to the experiment's purpose and the participating groups, scored the AUT responses individually according to criteria developed by the author based on existing literature [41–44]. To measure fluency, the coders counted the number of responses, excluding ordinary (e.g., building a wall with bricks) and infeasible (e.g., using bricks as food) uses, and calculated the total number of responses for the eight objects. For flexibility, the coders evaluated the number of categories into which the responses for each object could be divided (e.g., hitting with bricks and throwing bricks were evaluated as one category) and then calculated the total number of the eight objects. For originality, the coders rated how unusual each response was using a seven-point scale (1 = *usual* to 7 = *unusual*) and then calculated the mean of all responses. For elaboration, the coders rated the level of detail and the development of each response (e.g., using bricks "as a weight" is less detailed; "as a weight to hold down the four corners so that the ground sheet does not fly away" is more detailed) using a seven-point scale (1 = *not detailed* to 7 = *detailed*) and then calculated the mean of all responses. The intra-class correlations (ICCs), which indicate the consistency of interval measures across coders, for fluency and flexibility were .986 and .977, respectively. Kendall's coefficient of concordance (W), which indicates the consistency of ordinal measures across coders, for originality and

elaboration were .797 and .745, respectively. Since the values suggested sufficient consistency, the mean values of the three raters were used in subsequent analyses.

## Results

One-way analysis of variances were conducted to test for significant differences among the three groups (HMD vs. computer screen vs. control) in their fluency, flexibility, originality, and elaboration scores. Fig 2 shows boxplots, dot plots, and mean values for each score. The analysis showed significant differences between the groups for fluency ($F(2, 72) = 3.859$, $p = .026$, $\eta^2 = .097$, 95%CI [.000, .223]), flexibility ($F(2, 72) = 3.893$, $p = .025$, $\eta^2 = .098$, 95%CI [.000, .224]), and originality ($F(2, 72) = 8.016$, $p < .001$, $\eta^2 = .182$, 95%CI [.038, .321]). The difference in elaboration scores was not significant ($F(2, 72) = 0.433$, $p = .650$, $\eta^2 = .012$, 95%CI [.000, .080]). Multiple comparisons using the Holm method revealed that the fluency ($t(72) = 2.749$, adj.$p = .023$, $d = 0.765$, 95%CI [0.199, 1.332]), flexibility ($t(72) = 2.771$, adj.$p = .021$, $d = 0.771$, 95%CI [0.204, 1.338]), and originality scores ($t(72) = 3.980$, adj.$p < .001$, $d = 1.108$, 95%CI [0.519, 1.697]) were higher in the HMD group than in the computer screen group. The HMD group also recorded higher originality scores than the control group ($t(72) = 2.372$, adj.$p = .041$, $d = .660$, 95%CI [0.099, 1.222]). There was no significant difference in any of the scores between the computer screen group and the control group ($ts(72) < 1.608$, $ps > .112$, $ds < .448$).

## Discussion

Experiment 1, characterized by VR viewing medium, showed that the HMD group had higher fluency, originality, and flexibility scores than the computer screen group. This result supports Prediction 1, suggesting that VR experiences on an immersive HMD resulted in better divergent thinking performance. Similar to studies showing that high immersion in the natural environment facilitates divergent thinking in real-world settings (e.g., Lynch et al. [20]), the HMD's effect on facilitating psychological immersion and visual presence in VR [18, 24] may have promoted divergent thinking. However, this study did not measure the degree of immersion of the participants, and careful interpretation is needed to determine whether the differences between the groups were due to differences in immersion or other factors of the viewing medium.

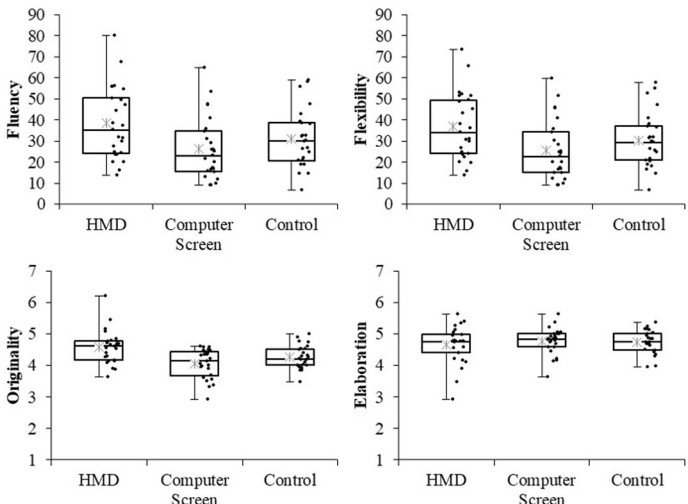

**Fig 2. Boxplots and dot plots for each score (Experiment 1).** Asterisks denote mean values.

Palanica et al. [23] conducted a comparison similar to the present study and reported the limited effect of the differences in the viewing medium on the AUT score. The difference in results between the present study and Palanica et al.'s [23] study may be the duration of video viewing. In the present study, the viewing time was eight minutes prior to the task and 16 minutes during the task. In Palanica et al.'s [23] study, the viewing time was 45 seconds prior to the task and three minutes during the task, which may have been too short an exposure to the VR environment. Therefore, the viewing duration may significantly impact the effects of the VR environment.

In Experiment 1, viewing the video with an HMD had the largest effect on originality, and the originality score of the HMD group was significantly different compared with both the computer screen group and the control group. This result suggests that the VR experience using immersive HMDs had a large effect, particularly on producing original ideas. Meanwhile, fluency and flexibility scores were not significantly different between the HMD and control groups. In Palanica et al.'s [23] study, the groups completing the AUT in real-world settings scored higher regardless of the location. Similarly, the control group in the present experiment was likely to achieve high scores because they looked around the actual laboratory, though it was visually closed and an environment less conducive to divergent thinking. It may be necessary to enhance the realisticness of the VR experience through sensory modalities beyond vision to promote more robust divergent thinking.

Although the HMD and computer screen groups differed in fluency, flexibility, and originality, their elaboration scores showed no differences. Given that elaboration measures the detail of responses, it tends to be a trade-off with fluency, which measures the number of responses [37]. Expressly, if the participants attempt to produce many responses within the time limit, the detail of each answer is likely to be low. However, in Experiment 1, despite the significantly higher fluency of the HMD group, elaboration was not significantly lower. Therefore, we can interpret this result positively: VR experience via an HMD generated numerous ideas without reducing the detail of the responses.

The researcher verbally asked the participants how often they used VR and found that only one participant used VR daily. Hence, the VR experience was relatively novel for many participants. This bias may have influenced the results, because some previous studies have suggested that novelty enhances creative performance and cognitive flexibility [45–47]. Expressly, it is unclear whether the high AUT score in the HMD group was due to an immersive viewing medium or the novelty of the VR experience. If the novel experience of VR itself affects divergent thinking, then any VR environment can be presumed to encourage divergent thinking. Clarifying this point requires a comparison of the effects of different VR environments on divergent thinking. Thus, in Experiment 2, I manipulated the spatial openness of the VR environment to test Prediction 2. Furthermore, I tested whether the results of Experiment 1 could be attributed to viewing a visually open coast in VR with the immersive HMD or the novelty of the VR experience by determining whether divergent thinking differs across varied VR environments.

## Experiment 2

Experiment 2 investigated the effect of the VR environment's spatial openness on divergent thinking. The AUT scores of the group viewing the 360˚ video of the coast on the HMD were compared with the scores of the group viewing the 360˚ video of the laboratory on the HMD.

### Methods

**Participants.** The participants were 50 university students (26 males; $M_{age}$ = 21.4 years, $SD_{age}$ = 2.6) who had not participated in Experiment 1. The sample size was the same as in

Experiment 1 to facilitate the comparison of results. Participants were recruited through on-campus advertisements and assigned to one of the two groups while ensuring equal gender ratios (coast and laboratory; $n = 25$). All participants provided their written informed consent prior to beginning the experiment and were compensated JPY 1,000 after the experiment for their participation. This experiment was approved by the ethics review committee of the institution at which the experiments were conducted.

**Stimuli and apparatus.** The stimuli were the same 360˚ video of the coast used in Experiment 1 (Fig 1A) and a video of the laboratory from Experiment 1 (Fig 1B) captured by a 360˚ camera (RICHO THETA V) from the center of the room. Participants viewed either of the videos on the same headset (Oculus Go) as in Experiment 1.

**Procedure.** The procedure was the same as that in Experiment 1. The coast group followed the same process as the HMD group in Experiment 1. The laboratory group also followed the same process; however, they viewed the laboratory video.

**AUT scoring.** The same three coders in Experiment 1 scored the AUT responses following the same criteria as Experiment 1. The ICCs for fluency and flexibility were .993 and .981, respectively. Kendall's W for originality and elaboration were .858 and .747, respectively.

## Results

I conducted $t$-tests to assess significant differences between the two groups (coast vs. laboratory) in fluency, flexibility, originality, and elaboration scores. Fig 3 shows boxplots, dot plots, and mean values for each score. The analysis revealed significant differences between the groups in fluency ($t(48) = 3.842$, $p < .001$, $d = 1.070$, 95%CI [0.484, 1.656]), flexibility ($t(48) = 3.934$, $p < .001$, $d = 1.095$, 95%CI [0.507, 1.683]), and originality ($t(48) = 3.351$, $p = .002$, $d = 0.933$, 95%CI [0.356, 1.510]). The difference in elaboration scores was not significant ($t(48) = 0.337$, $p = .737$, $d = 0.094$, 95%CI [-0.640, 0.452]).

## Discussion

In Experiment 2, which manipulated spatial openness, the coast group recorded higher fluency, originality, and flexibility scores than the laboratory group. This result supports Prediction 2, suggesting that a visually open VR environment elicits better divergent thinking. In

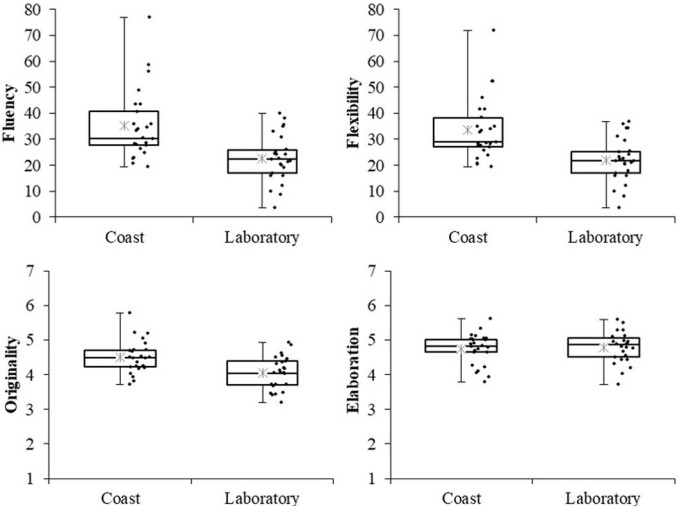

**Fig 3. Boxplots and dot plots for each score (Experiment 2).** Asterisks denote mean values.

particular, the effect sizes of fluency and flexibility scores were larger than those in Experiment 1, suggesting that spatial openness may strongly influence the production of diverse ideas. Meanwhile, the elaboration scores between the conditions showed no significant difference, as in Experiment 1. This finding can be interpreted favorably as participants in the coast group produced more ideas without compromising detail in their responses.

Previous research in real-world settings has shown that rooms that activate free concepts, such as broad rooms, high ceilings, or disorderly rooms, promote divergent thinking [21, 25, 29]. Similarly, in the present study, the VR environment of the visually open coast with no ceiling and an unobstructed view resulted in higher AUT scores because it activated the concept of freedom compared with the visually closed laboratory surrounded by ceiling and walls.

In Experiment 1, I considered the possibility that the novelty of the VR experience, rather than the VR visually open coast on an immersive HMD, promoted divergent thinking. If the novelty of the VR experience facilitated divergent thinking, the AUT scores in Experiment 2 would be high regardless of the VR environment. However, viewing the laboratory video, a visually closed environment, did not result in AUT scores as high as those viewing the coast video. Furthermore, comparing the AUT scores of the control group in Experiment 1, which viewed the actual laboratory, with the scores of the laboratory group in Experiment 2, which viewed a 360˚ video of the laboratory on an HMD, revealed that the control group had higher scores in fluency ($t(48)$ = 2.468, $p$ = .017, $d$ = 0.687, 95%CI [0.124, 1.250]) and flexibility ($t(48)$ = 2.598, $p$ = .012, $d$ = 0.723, 95%CI [0.159, 1.288]). Moreover, there were no differences in originality ($t(48)$ = 1.674, $p$ = .101, $d$ = 0.466, 95%CI [-0.088, 1.020]) or elaboration ($t(48)$ = 1.036, $p$ = .305, $d$ = 0.289, 95%CI [-0.260, 0.837]). Expressly, there was no evidence that experiencing a visually closed environment in VR enhances divergent thinking when compared to experiencing the same environment in a real-world setting. Therefore, I can conclude that the results of Experiment 1 were not due to the novelty of the VR experience.

## General discussion

This study examined the effects of a VR experience on divergent thinking. Experiments 1 and 2 confirmed that viewing a visually open VR environment on an HMD leads to higher scores in fluency, flexibility, and originality of ideas without sacrificing elaboration. The effect size (Cohen's $d$) suggested that the viewing medium had a large effect on originality, whereas spatial openness had a large effect on fluency and flexibility.

Studies conducted in real-world settings have shown that immersion in natural environments and environments that activate the concept of freedom promote divergent thinking [20–22, 25, 27, 29, 30]. The present study confirmed similar effects in VR by manipulating the VR viewing medium, which yields different levels of psychological immersion and senses of visual presence, and the spatial openness of the VR environment. These results suggest that the findings observed in real-world settings are applicable to VR.

## Limitations and future prospects

This study compared conditions that are thought to have markedly different effects on divergent thinking. Therefore, several issues arise, and further studies are needed to conclude that this study's results are due to immersion and openness of space.

This study has a few limitations. First, there is the issue of confounding factors. In this study, I manipulated the medium of the 360˚ video presentation (HMD vs. computer screen). However, we cannot deny the possibility that posture (standing vs. sitting) and viewpoint manipulation (moving the head vs. moving the mouse) may have affected the results. An approach using a mobile tablet instead of a computer screen should be devised, as in Palanica

et al.'s [23] study, allowing the participants to change their viewpoint by moving their heads while standing.

Regarding the manipulation of spatial openness (coast vs. laboratory), there are also confounding factors, such as environment (nature vs. laboratory), novelty (coast video viewed for the first time vs. laboratory video of participants' present setting), number of objects (few vs. many), and lighting (natural light vs. fluorescent light). Previous studies have suggested the association between these factors and creative thinking (e.g., [21, 30, 45, 48]), and the present results may reflect the effects of multiple factors. Further validation is needed using stimuli designed to control these factors.

The second limitation is a lack of examination of the impact of immersion. This experiment showed the superiority of immersive HMDs compared to computer screens. However, the impact compared to real-world settings is insufficient because a real-world coast setting was not provided. To verify this, we need to provide a real-world open environment condition.

Additionally, measuring immersion or sense of presence using questionnaires or physiological indices would be helpful (Grassini and Laumann [49] provided a systematic review of the measures). Based on previous studies [18, 24], this study assumes that HMDs provide more immersion in the VR environment than computer screens. Nevertheless, it was not possible to confirm whether Experiment 1's HMD group was more immersed in the VR environment than the computer screen group. A manipulation check using the indices of immersion is needed. Such a manipulation check would allow for the possibility to examine the relationship between the degree of immersion and AUT scores by analyzing whether the variation in AUT scores can be explained by individual differences in immersion.

Furthermore, it is interesting to focus on multiple senses. Although this study manipulated only visual information, immersion is often discussed in terms of the number of sensory modalities available to the user [24]. The present study and Palanica et al. 's [23] study show that environmental differences affect divergent thinking in VR settings, as shown in real-world settings. However, neither has elicited a higher divergent thinking performance in the VR environment compared to a real-world environment. This may be because the number and quality of sensory modalities reproduced via HMDs are inferior to those in real-world environments. Technically, some immersive VR systems can provide sensory modalities beyond the visual. When further technological development enables VR systems to reproduce more realistic sensations, future research should explore whether VR environments can improve divergent thinking as much as or more than real-world environments.

The third issue to examine in future research is the influence of habituation on the VR environment, such as viewing time and frequency of daily use. In the present study, the difference in AUT scores between the 2D video and 3D VR conditions was larger than that in Palanica et al.'s [23] study. This difference could be attributed to the longer video viewing time in the present study, which resulted in a higher level of immersion in the VR environment. Notably, neither the present study nor Palanica et al. [23] found AUT scores to be higher in VR environments, which are thought to promote divergent thinking, than in real-world environments, which are thought to be less likely to promote divergent thinking. Participants' limited use of VR on a daily basis may explain the lack of support for this expectation. As mentioned, the researcher orally asked the participants about their daily VR experience after the experiment and found that only one participant used VR daily. VR technology remains in development and is gradually becoming more accessible to the public. It will be interesting to examine VR's effect on divergent thinking as more people become familiar with VR.

Thus, although this study has many limitations, it presents the possibility that VR environments can affect divergent thinking. There have been studies on the development of VR technologies and training programs for creative activities [8–10], but these do not consider VR

environments' ability to enhance creative thinking. By overcoming the above limitations and accumulating knowledge of VR environments, we can expect more efficient creative activities in VR, such as interaction with the VR environment. This study can serve as a guideline for future research in this area.

## Supporting information

**S1 Data.**
(XLSX)

## Author Contributions

**Conceptualization:** Kenshiro Ichimura.

**Data curation:** Kenshiro Ichimura.

**Formal analysis:** Kenshiro Ichimura.

**Funding acquisition:** Kenshiro Ichimura.

**Investigation:** Kenshiro Ichimura.

**Methodology:** Kenshiro Ichimura.

**Project administration:** Kenshiro Ichimura.

**Resources:** Kenshiro Ichimura.

**Software:** Kenshiro Ichimura.

**Supervision:** Kenshiro Ichimura.

**Validation:** Kenshiro Ichimura.

**Visualization:** Kenshiro Ichimura.

**Writing – original draft:** Kenshiro Ichimura.

**Writing – review & editing:** Kenshiro Ichimura.

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
