## [Decision Letter · Decision Letter 0]

14 Nov 2022

PONE-D-22-21099Effects of immersion and spatial openness of the virtual reality environment on divergent thinkingPLOS ONE

Dear Dr. Ichimura,

Thank you for submitting your manuscript to PLOS ONE. After careful consideration, we feel that it has merit but does not fully meet PLOS ONE’s publication criteria as it currently stands. Therefore, we invite you to submit a revised version of the manuscript that addresses the points raised during the review process.

Two Reviewers have evaluated the manuscript. Although Reviewer 2 gave a positive opinion and suggestions for improvement of the take-home message of the work, Reviewer 1 found several possible methodological issues. I encourage Authors to submit a revised version taking into account all Reviewers' concerns, especially in terms of methodological clarity, correct identification of the variables involved in the study, possible adjustments to data analysis and discussion/limitations. 

We look forward to receiving your revised manuscript.

Kind regards,

Stefano Triberti, Ph.D.

Academic Editor

PLOS ONE

Journal Requirements:

Reviewers' comments:

Reviewer's Responses to Questions

**Comments to the Author**

1. Is the manuscript technically sound, and do the data support the conclusions?

Reviewer #1: Partly

Reviewer #2: Yes

2. Has the statistical analysis been performed appropriately and rigorously? 

Reviewer #1: Yes

Reviewer #2: Yes

3. Have the authors made all data underlying the findings in their manuscript fully available?

Reviewer #1: Yes

Reviewer #2: Yes

4. Is the manuscript presented in an intelligible fashion and written in standard English?

Reviewer #1: Yes

Reviewer #2: Yes

5. Review Comments to the Author

Reviewer #1: - The work explores the connection between immersion and divergent thinking with the help of VR. 2 experiments were conducted; 1) comparing the effect of different level of immersion on divergent thinking, and 2) comparing open vs closed environment when viewed in VR. The authors claim that higher immersion leads to more divergent thinking and that VR novelty effect is not a factor driving this increase in divergent thinking behavior. Although the experiments were conducted well, I do not believe that the author’s claim are well supported as there are many open questions and missing data points. Please seem my detailed comments below.

- “The present study focused on divergent thinking because of its affinity with VR” – This statement needs to be expanded on and would benefit from a citation. The connection here is not evident. The following text in the same paragraph says that other studies used this, but it does not explain the why. Considering this work is focusing on divergent thinking, it needs to do a better job of explaining the why behind the focus.

- “Researchers have examined the effects of immersion on creativity in real-world 75 and VR settings [17,21–23].” – Similar to my previous comment, simply stating something and providing citations is not enough. The why of the investigation in immersion and divergent thinking is critical for the reader to understand but is missing here.

- The related work covers some good works, but does not discuss them well enough to show where this work fits in well. It would be ideal to discuss inferences and not just state results from a particular work.

- Experiment 1

o The author talks about the effects of immersion on divergent thinking, but they do not seem to have measured immersion using a survey or otherwise. They seemed to have only measured AUT and compared that. It is extremely critical to measure the levels of immersion in a study where immersion is an IV. Users can have varying levels of immersion even in VR and the high variability in the current AUT scores as can be seen in the box plot could have easily been because of the highly varied levels of immersion in VR. If the author did not measure these scores this should be listed as a limitation and perhaps can phrase their comparison as one between different levels of exposure rather than levels of immersion.

o Based on the fact that immersion scores were not measured some of the inferences made in the discussion are questionable. For example, “suggesting that high immersion in the environment through VR resulted in better divergent thinking performance” – We do not know this because immersion was not measured post experience.

- Experiment 2

o “The sample size was determined such that the number of participants in each group would be the same as that in Experiment 1” – Why was this a necessity? I would have likes to see the power analysis details similar to experiment 1.

o “I can conclude that the results of Experiment 1 were not because of the novelty of the VR experience.” – This conclusion cannot be drawn without comparing the scores of the control group who saw the lab in experiment 1 to the ones who viewed a 360 video of the lab in VR. Please revise.

o The narrative of this experiment is a bit more digestible as it does not directly talk about immersion and its connection to divergent thinking. However, immersion scores would have immensely helped explain the high variance in the coast 360 video condition. The author seems to have completely overlooked this aspect and I would like some discussion around this to be included in the text.

- The author seems to often compare their findings to works conducted in the real world. However, there is a drastic difference between the two, i.e. the combined effect of multiple senses. Prior research shows that engaging other senses greatly enhances immersion and presence. As the author pointed out, they only engaged the visual sense in their experiments here. Although the author does discuss this in the limitations, this discrepancy and potential effects of the same need to be discussed in detail as part of the discussion.

Reviewer #2: The submitted article is particularly interesting and crucial in terms of designing virtual reality experiences. The literature cited is carefully chosen and relevant to the two studies. However, I recommend adding two or three sentences in the conclusions to provide general indications on how the results of the two studies (particularly the second on spatial openness) can be used in the design of virtual reality, training, or online training experiences. Indeed, I believe that this would add value to the article and provide a good starting point for future research. I also suggest to change a letter in line 27 ("Virtual reality" (VR) by inserting the capital r), and to line 37 (where it is written "special openess" instead of spatial).

6. PLOS authors have the option to publish the peer review history of their article (what does this mean?). If published, this will include your full peer review and any attached files.

Reviewer #1: No

Reviewer #2: No

---

## [Author Response · Author response to Decision Letter 0]

10 Jan 2023

Dear Editor and Reviewers,

Thank you very much for the careful review of my paper. I greatly appreciate your valuable comments. I have revised the paper accordingly, and my responses to your comments are presented below. Additionally, I have made many corrections to the language. I appreciate your consideration.

Comment 1 from Reviewer #1

“The present study focused on divergent thinking because of its affinity with VR” – This statement needs to be expanded on and would benefit from a citation. The connection here is not evident. The following text in the same paragraph says that other studies used this, but it does not explain the why. Considering this work is focusing on divergent thinking, it needs to do a better job of explaining the why behind the focus.

Response to comment 1

I have revised the description in the relevant paragraphs to clarify the reasons and significance of the focus on VR and divergent thinking.

(line 61 – 72)

Comment 2 from Reviewer #1

“Researchers have examined the effects of immersion on creativity in real-world 75 and VR settings [17,21–23].” – Similar to my previous comment, simply stating something and providing citations is not enough. The why of the investigation in immersion and divergent thinking is critical for the reader to understand but is missing here.

Response to comment 2

In the paragraph preceding the section in question, I have added an explanation to clarify the reasons for investigating immersion and divergent thinking in VR settings.

(line 86 – 97)

Comment 3 from Reviewer #1

The related work covers some good works, but does not discuss them well enough to show where this work fits in well. It would be ideal to discuss inferences and not just state results from a particular work.

Response to comment 3

I assumed this comment was in response to the first paragraph of the “Environment and divergent thinking” section. The remaining paragraphs in this section provide inferences from the mentioned studies. Additionally, I have added an explanation of the significance of examining the impact of environmental differences in a VR setting.

(line 135 – line 138)

Comment 4 from Reviewer #1

The author talks about the effects of immersion on divergent thinking, but they do not seem to have measured immersion using a survey or otherwise. They seemed to have only measured AUT and compared that. It is extremely critical to measure the levels of immersion in a study where immersion is an IV. Users can have varying levels of immersion even in VR and the high variability in the current AUT scores as can be seen in the box plot could have easily been because of the highly varied levels of immersion in VR. If the author did not measure these scores this should be listed as a limitation and perhaps can phrase their comparison as one between different levels of exposure rather than levels of immersion.

Response to comment 4

In response to your suggestion, I have revised the phrasing throughout the paper to ensure sentence clarity. Similarly, the title and abstract have been revised. In addition, I have revised the “General discussion” section to provide a more detailed explanation of this limitation (line 425 – 434) and have added an explanation to the “Discussion” section of Experiment 1 (line 270 – 272). 

Comment 5 from Reviewer #1

Based on the fact that immersion scores were not measured some of the inferences made in the discussion are questionable. For example, “suggesting that high immersion in the environment through VR resulted in better divergent thinking performance” – We do not know this because immersion was not measured post experience.

Response to comment 5

Per the response to comment 4, I have revised the phrasing throughout the manuscript to avoid misinterpretation.

Comment 6 from Reviewer #1

“The sample size was determined such that the number of participants in each group would be the same as that in Experiment 1” – Why was this a necessity? I would have likes to see the power analysis details similar to experiment 1.

Response to comment 6

To align the precision of the interval estimation and to facilitate a comparison between Experiment 1 and Experiment 2, the sample size in Experiment 2 was the same as in Experiment 1. This procedure was approved during the ethics review. As you pointed out, conducting another power analysis might have been effective, but since I cannot make any changes now, I have left the statements as they are.

Comment 7 from Reviewer #1

“I can conclude that the results of Experiment 1 were not because of the novelty of the VR experience.” – This conclusion cannot be drawn without comparing the scores of the control group who saw the lab in experiment 1 to the ones who viewed a 360 video of the lab in VR. Please revise.

Response to comment 7

If the results of Experiment 1 were due to the novelty of the VR experience, we would expect no difference between the coast group and the laboratory group in Experiment 2 because both experienced VR. However, significant differences were found between the groups, leading to this conclusion.

Regarding your point, since the mean AUT scores of the laboratory group in Experiment 2 is lower than the control group in Experiment 1, I conclude that the effect of the VR experience alone on the improvement of the AUT scores is small. I have added this point to the paper as it reinforces the conclusion.

(line 377 – 386)

Comment 8 from Reviewer #1

The narrative of this experiment is a bit more digestible as it does not directly talk about immersion and its connection to divergent thinking. However, immersion scores would have immensely helped explain the high variance in the coast 360 video condition. The author seems to have completely overlooked this aspect and I would like some discussion around this to be included in the text.

Response to comment 8

Per the response to comment 4, I have revised the explanation in the “General discussion” section to include more detail.

(line 425 – 434)

Comment 9 from Reviewer #1

- The author seems to often compare their findings to works conducted in the real world. However, there is a drastic difference between the two, i.e. the combined effect of multiple senses. Prior research shows that engaging other senses greatly enhances immersion and presence. As the author pointed out, they only engaged the visual sense in their experiments here. Although the author does discuss this in the limitations, this discrepancy and potential effects of the same need to be discussed in detail as part of the discussion.

Response to comment 9

The fact that no differences were found between VR and real space may be because of the differences you have pointed out. I have added an explanation of this point to the “General discussion” section.

(line 437 – 446)

Comment 1 from Reviewer #2

The submitted article is particularly interesting and crucial in terms of designing virtual reality experiences. The literature cited is carefully chosen and relevant to the two studies. However, I recommend adding two or three sentences in the conclusions to provide general indications on how the results of the two studies (particularly the second on spatial openness) can be used in the design of virtual reality, training, or online training experiences. Indeed, I believe that this would add value to the article and provide a good starting point for future research.

Response to comment 1

Because of the limitations mentioned in the “General discussion” section, I believe that it is difficult to refer to the study results’ applicability. Therefore, I have provided a brief explanation of the implied implications at the end of the paper to avoid exaggerations.

(line 462 – 467)

Comment 2 from Reviewer #2

I also suggest to change a letter in line 27 (“Virtual reality” (VR) by inserting the capital r), and to line 37 (where it is written “special openess” instead of spatial).

Response to comment 2

Thank you for your suggestion. I have made the necessary corrections per your comments.

(line 26 and 37)

---

## [Decision Letter · Decision Letter 1]

13 Mar 2023

Effects of Virtual Reality’s viewing medium and the environment’s spatial openness on divergent thinking

PONE-D-22-21099R1

Dear Dr. Ichimura,

We’re pleased to inform you that your manuscript has been judged scientifically suitable for publication and will be formally accepted for publication once it meets all outstanding technical requirements.

Kind regards,

Stefano Triberti, Ph.D.

Academic Editor

PLOS ONE

Additional Editor Comments (optional):

Reviewers' comments:

Reviewer's Responses to Questions

**Comments to the Author**

1. If the authors have adequately addressed your comments raised in a previous round of review and you feel that this manuscript is now acceptable for publication, you may indicate that here to bypass the “Comments to the Author” section, enter your conflict of interest statement in the “Confidential to Editor” section, and submit your "Accept" recommendation.

Reviewer #1: All comments have been addressed

2. Is the manuscript technically sound, and do the data support the conclusions?

Reviewer #1: Yes

3. Has the statistical analysis been performed appropriately and rigorously? 

Reviewer #1: Yes

4. Have the authors made all data underlying the findings in their manuscript fully available?

Reviewer #1: Yes

5. Is the manuscript presented in an intelligible fashion and written in standard English?

Reviewer #1: Yes

6. Review Comments to the Author

Reviewer #1: (No Response)

7. PLOS authors have the option to publish the peer review history of their article (what does this mean?). If published, this will include your full peer review and any attached files.

Reviewer #1: No

---

## [Editor Report · Acceptance letter]

16 Mar 2023

PONE-D-22-21099R1 

Effects of Virtual Reality’s viewing medium and the environment’s spatial openness on divergent thinking 

Dear Dr. Ichimura:

I'm pleased to inform you that your manuscript has been deemed suitable for publication in PLOS ONE. Congratulations! Your manuscript is now with our production department. 

Kind regards, 

on behalf of

Dr. Stefano Triberti 

Academic Editor

PLOS ONE